# Low Kruskal-Rank Adaptation

**Yixing Xu** [1]  **Guanchen Li** [1]  **Chao Li** [1]  **Xuanwu Yin** [1]  **Dong Li** [1]  **Spandan Tiwari** [1]  **Ashish Sirasao** [1]  **Emad Barsoum** [1]

## Abstract

Low-rank adaptation (LoRA) is one of the most widely used parameter-efficient fine-tuning (PEFT) methods for adapting pre-trained large language models (LLMs) to downstream tasks. Although LoRA significantly reduces the number of trainable parameters and lowers fine-tuning costs, its performance is often limited by the inherent low-rank assumption. In this paper, we revisit the notion of rank for LoRA update matrices and show that the standard matrix rank fails to capture duplicated directions and redundancy in the update subspace. Motivated by this analysis, we argue that the Kruskal rank offers a more informative criterion for characterizing update diversity. We therefore propose **Low Kruskal Rank Adaptation** (LoKRA), a new PEFT algorithm with provable theoretical guarantees that mitigates the limitations of LoRA. We further introduce LoKRA$^+$, an enhanced variant that provides a tighter theoretical lower bound on the Kruskal rank and yields stronger empirical performance. Experiments on multiple LLMs show that our approach consistently outperforms LoRA and other baselines, establishing state-of-the-art performance across a range of benchmarks. The code can be viewed at GitHub.

## 1. Introduction

Large language models (LLMs) have demonstrated strong performance across a wide range of natural language processing (NLP) tasks, driven by their massive parameter counts and training on extensive datasets with substantial computational resources (Dubey et al., 2024; Guo et al., 2025; Yang et al., 2025a; Xu et al., 2025b). However, fully fine-tuning (FFT) such models for each downstream task is prohibitively expensive in both computation and memory. To mitigate these costs, parameter-efficient fine-tuning (PEFT) methods have been developed to significantly reduce the number of trainable parameters while preserving task performance (Houlsby et al., 2019; Lester et al., 2021; Li & Liang, 2021; Xu et al., 2025a). Low-rank adaptation (LoRA) (Hu et al., 2022) has emerged as one of the most widely adopted techniques due to its simplicity and practicality. It leaves the inference-time architecture unchanged and achieves strong empirical performance by training only a small fraction of additional parameters.

The standard LoRA method (Hu et al., 2022) assumes that the parameter update matrix processes a low-rank characteristic, therefore reducing the number of trainable parameters by decomposing the large update matrix $\Delta W$ into two smaller matrices $BA$. Subsequent work has mainly pursued two directions to improve LoRA. The first is to modify the parameterization of the update matrix. DoRA (Liu et al., 2024) decomposes the pre-trained weight into magnitude and direction for fine-tuning. HiRA (Huang et al., 2025) uses a Hadamard product to retain high-rank update parameters and improve the model capacity. Kron-LoRA (Shen, 2025) replaces matrix multiplication with the Kronecker product to theoretically derive a high rank update matrix. Another family of methods focuses on improving memory efficiency through gradient-space optimization. For example, GaLore (Zhao et al., 2024) is a gradient low-rank projection strategy that projects full-parameter gradients onto a low-rank subspace during optimization, enabling memory-efficient training while retaining full-parameter updates.

A large body of existing methods seeks to increase the rank of the update matrix by modifying either the forward or backward propagation passes. While they typically yield a higher rank than vanilla LoRA, they share a fundamental limitation: under a constrained parameter budget, standard matrix rank is too coarse a measure to characterize the expressive capacity of LoRA update matrices.

Therefore, in this paper, we revisit the rank in LoRA updates and show that the standard matrix rank is insufficient to identify redundancy and duplicated directions in parameter updates. In contrast, Kruskal rank provides a more appropriate criterion: it captures *global and robust* indepen-

---

[1]Advanced Micro Devices, Inc., Beijing, China. Correspondence to: Yixing Xu <yixingx@amd.com>.

*Proceedings of the 43rd International Conference on Machine Learning*, Seoul, South Korea. PMLR 306, 2026. Copyright 2026 by the author(s).

dence by requiring every subset of $k$ columns to be linearly independent, thereby reflecting the intrinsic independence structure of the update matrix's column (or row) subspace.

Based on the analysis above, we propose Low Kruskal rank adaptation (LoKRA), a new PEFT algorithm that promotes a higher Kruskal rank for the update matrix with theoretical guarantees, thereby ensuring minimal redundancy under limited trainable parameters. Specifically, we maximize the log-determinant of each sub-matrix to promote linear independence. This objective is further refined to reduce the training overhead while preserving its effectiveness. In addition, we study an update parameterization that is more naturally aligned with Kruskal rank, and propose LoKRA$^+$, which replaces standard matrix multiplication with the Khatri–Rao product. This enhanced variant provides a tighter theoretical lower bound on the Kruskal rank and delivers stronger empirical performance.

Our experimental results demonstrate the effectiveness of optimizing the Kruskal rank. Compared to the original LoRA, LoKRA achieves a maximum accuracy improvement of **5.0%** on commonsense reasoning benchmarks with LLaMA3-8B and **3.7%** on mathematical reasoning datasets with Deepseek-R1-Distilled-Qwen-1.5B. Moreover, LoKRA consistently outperforms prior state-of-the-art methods, such as LoRA$^+$, DoRA, and HiRA. LoKRA$^+$ further improves upon LoKRA and substantially narrows the performance gap between the LoRA-style PEFT methods and full fine-tuning.

## 2. Related Work

**LoRA-based Parameter-Efficient Fine-Tuning**. Early parameter-efficient fine-tuning (PEFT) for LLMs inserts small task-specific modules into a frozen backbone, such as adapter layers with bottleneck MLP blocks (Houlsby et al., 2019), learnable continuous prompts at the input (Lester et al., 2021), and attention-prefix level insertion (Li & Liang, 2021). Low-Rank Adaptation (LoRA) (Hu et al., 2022) instead injects trainable low-rank updates ($\Delta \boldsymbol{W} = \boldsymbol{B}\boldsymbol{A}$) into existing weight matrices while keeping the backbone frozen, achieving strong empirical performance with only a small fraction of additional parameters.

Beyond this basic formulation, a large body of work modifies the parameterization or forward/backward propagation of LoRA-style adapters. LoRA+ (Hayou et al., 2024) assigns different learning rates to the $\boldsymbol{A}$ and $\boldsymbol{B}$ matrices, improving optimization efficiency for wide models. AdaLoRA (Zhang et al., 2023) re-parameterizes the update via singular value decomposition and dynamically reallocates rank budgets across layers according to estimated importance. VeRA (Kopiczko et al.) further reduces trainable parameters by sharing frozen random low-rank ma-

trices across layers and learning only small per-layer vectors. DoRA (Liu et al., 2024) decomposes each weight into magnitude and direction and applies low-rank updates only to the directional component, aiming to improve stability and expressivity over standard additive low-rank updates. HydraLoRA (Tian et al., 2024) adopts an asymmetric architecture with a shared $\boldsymbol{A}$ and multiple task- or component-specific $\boldsymbol{B}$ branches, improving parameter efficiency and adaptability. GaLore (Zhao et al., 2024) reduces fine-tuning memory and compute by updating gradients in a learned low-rank subspace instead of the full parameter space.

**Towards High-Rank Parameter-Efficient Fine-Tuning**. The strict low-rank constraint in standard LoRA limits the expressivity of its updates and can leave a noticeable gap to full fine-tuning on challenging tasks. To mitigate this, HiRA (Huang et al., 2025) replaces the pure low-rank additive update with a Hadamard modulation of the frozen weights, effectively yielding high-rank behavior at LoRA-level parameter cost. Kron-LoRA (Shen, 2025) instead parameterizes the adapter with a Kronecker product to obtain a richer update structure per parameter than a simple rank factorization. RandLoRA (Albert et al.) forms full-rank updates as learned combinations of multiple fixed low-rank random bases, narrowing the gap to full fine-tuning under a parameter budget.

Overall, these methods modify the underlying matrix operation (Hadamard product, Kronecker product, and random bases) to increase the effective rank of $\Delta \boldsymbol{W}$ and partially alleviate LoRA's low-rank limitation. In contrast, our method directly promotes a higher Kruskal rank for $\Delta \boldsymbol{W}$, enforcing stronger directional independence than standard rank-based criteria and bringing the update closer to full-parameter fine-tuning.

## 3. Preliminaries

In this section, we first introduce the preliminaries of LoRA, the standard matrix rank, and the Kruskal rank (shortened to K-rank). Then, we provide a brief analysis demonstrating that LoRA can benefit more from the K-rank than from the standard matrix rank.

### 3.1. Low-Rank Adaptation (LoRA)

Given an input feature $\boldsymbol{x} \in \mathbb{R}^{c_{in} \times \text{batch}}$ and a pretrained weight matrix $\boldsymbol{W} \in \mathbb{R}^{c_{out} \times c_{in}}$, LoRA improves downstream performance by introducing a low-rank update $\Delta \boldsymbol{W} \in \mathbb{R}^{c_{out} \times c_{in}}$, parameterized as $\Delta \boldsymbol{W} = \boldsymbol{B}\boldsymbol{A}$ with $\boldsymbol{A} \in \mathbb{R}^{r \times c_{in}}$ and $\boldsymbol{B} \in \mathbb{R}^{c_{out} \times r}$, where $r$ is a rank hyperparameter. During fine-tuning, LoRA freezes $\boldsymbol{W}$ and trains only $\boldsymbol{A}$ and $\boldsymbol{B}$, the forward pass can be written as:

$$f(\boldsymbol{x}) = (\boldsymbol{W} + \Delta \boldsymbol{W})\boldsymbol{x} = \boldsymbol{W}\boldsymbol{x} + \boldsymbol{B}\boldsymbol{A}\boldsymbol{x}. \quad (1)$$

Given $r \ll \min(c_{in}, c_{out})$, LoRA can largely reduce the trainable parameters from $\mathcal{O}(c_{in} \times c_{out})$ to $\mathcal{O}(c_{in} + c_{out})$.

## 3.2. Matrix Rank and Kruskal Rank

First, we give the definition of matrix rank (Frobenius, 1878) and Kruskal rank (Kruskal, 1977) of a given matrix.

**Definition 3.1** (Matrix rank). Let $W \in \mathbb{R}^{c_{out} \times c_{in}}$ be a matrix over a field $\mathbb{F}$, which equivalently defines a linear map $W : \mathbb{F}^{c_{in}} \rightarrowtail \mathbb{F}^{c_{out}}, x \mapsto Wx$. Then, the rank of the matrix $W$ is defined as:

$$\mathrm{rank}(W) = \dim_{\mathbb{F}}(\mathrm{Im}(W)), \qquad (2)$$

where $\mathrm{Im}(W) = \{Wx \mid x \in \mathbb{F}^{c_{in}}\} \subseteq \mathbb{F}^{c_{out}}$, $\dim_{\mathbb{F}}$ stands for the dimension of a vector space over the field $\mathbb{F}$.

Intuitively speaking, $\mathrm{rank}(W) = r$ indicates that there *exists* a subset of $r$ columns that are linearly independent, and no subset of size $r + 1$ is independent.

**Definition 3.2** (Kruskal rank). Let $W \in \mathbb{R}^{c_{out} \times c_{in}}$ be a matrix over a field $\mathbb{F}$. Let $W_{:,J_k}$ denote the submatrix of $W$ consisting of $k$ columns indexed by a subset $J_k \subseteq \{1, 2, \ldots, c_{in}\}$. The Kruskal rank of the matrix $W$ is:

$$\mathrm{kr}(W) = \max\{k \in \mathbb{N} \mid \forall J_k, \mathrm{rank}(W_{:,J_k}) = k\}. \quad (3)$$

Therefore, $\mathrm{kr}(W) = k$ indicates that *every* subset of $k$ columns is linearly independent, and no subset of size $k + 1$ is independent.

These definitions indicate that standard matrix rank is a *local* measure of independence: it only requires the **existence** of some linearly independent channels, while allowing arbitrary dependencies among the remaining channels. In the LoRA setting, rank guarantees at most the existence of $r$ independent update directions, whereas the other directions can still be highly correlated or even redundant. When such directions collapse or become degenerate, the expressive capacity of $\Delta W$ may deteriorate substantially. Unlike standard matrix rank, Kruskal rank captures *global* and robust independence by requiring **every** subset of up to $k$ columns (rows) to be linearly independent, thereby reflecting the overall diversity of the matrix's column (row) space. In the LoRA setting, a high Kruskal rank of the update matrix $\Delta W$ guarantees that any collection of up to $k$ update directions remains independent, allowing $\Delta W$ to adapt along diverse directions and potentially improving stability and generalization.

For example, consider a matrix with rank $r$ and Kruskal rank $k$. If we append a new column that is identical to an existing one, no new information is introduced. Nevertheless, the matrix rank remains $r$, whereas the Kruskal rank drops from $k$ to 1. This demonstrates that even when the update matrix in LoRA attains full matrix rank under its parameter budget

(*i.e.,* $r \ll \min(c_{in}, c_{out})$), it may still exhibit substantial redundancy and duplicated directions, which Kruskal rank explicitly exposes. In the following, we use rank to denote matrix rank and K-rank to denote Kruskal rank.

## 4. Method

In this section, we first give an introduction to the column and row K-rank (see Section 4.1). Then, we give a theoretical analysis of the K-rank of the update matrix $\Delta W$, and propose our LoKRA algorithm to increase the K-rank of learnable matrices $A$ and $B^\top$ during LoRA training based on the previous analysis (see Section 4.2). Finally, an enhanced LoKRA$^+$ algorithm is proposed, where the matrix multiplication in Equation (1) is replaced by a Khatri–Rao product to obtain a tighter theoretical lower bound on the K-rank of the update matrix (see Section 4.3).

### 4.1. Column K-rank and Row K-rank

Although the column rank and the row rank coincide for any matrix, this symmetry does not generally hold for the K-rank. Consequently, a natural question arises: should the optimization focus on the column K-rank or the row K-rank of the update matrix $\Delta W$?

Intuitively, a higher column K-rank indicates that any $k$ input dimensions are unlikely to induce linearly dependent effects on the output. The column K-rank, therefore, characterizes the model's capacity to preserve and discriminate information on the input side. A high column K-rank implies that even when only a small subset of input dimensions is active, the resulting output remains sufficiently informative to reliably distinguish among different input patterns.

A higher row K-rank indicates that the weight vectors associated with any $k$ output dimensions exhibit a low likelihood of linear dependence. Conceptually, the row K-rank characterizes the diversity and disentanglement capacity of the output representation: a high row K-rank implies that the output space admits a greater number of mutually independent directions, thereby enabling richer and more distinguishable output signals.

Thus, in the following, we propose LoKRA to optimize both column K-rank and row K-rank of the update matrix. We denote the K-rank as the column K-rank by default unless specified. More ablations are shown in Appendix F.

### 4.2. Low Kruskal Rank Adaptation (LoKRA)

**Analysis of K-rank.** We first analyze the K-rank properties of the update matrix $\Delta W$. By Definition 3.2, if $\Delta W$ contains an all-zero column (row), then its K-rank (row K-rank) is zero. Therefore, throughout the following analysis, we assume that $\Delta W$ has no all-zero columns or rows, and

we derive upper and lower bounds on the K-rank and row K-rank of $\Delta W = BA$.

**Theorem 4.1.** *Given two matrices $A \in \mathbb{R}^{r \times c_{in}}$ and $B \in \mathbb{R}^{c_{out} \times r}$, the K-rank and row K-rank of the matrix multiplication result is bounded by:*

$$1 \leq \text{kr}(BA) \leq \min\{\text{kr}(A), \text{rank}(B)\} \leq r, \quad (4)$$

$$1 \leq \text{kr}_{\text{row}}(BA) \leq \min\{\text{kr}(B^\top), \text{rank}(A)\} \leq r. \quad (5)$$

The proof is shown in Appendix A.

Note that in the LoRA setting, since $r \ll \min(c_{in}, c_{out})$, the factor matrices $A$ and $B$ can easily achieve full rank (Huang et al., 2025; Liu et al., 2024). Therefore, the column K-rank and row K-rank of $\Delta W$ are upper-bounded by the K-rank of $A$ and $B^\top$, respectively.

**Optimizing the K-rank of $A$ and $B^\top$.** Recall that Definition 3.2 implies that when $\text{kr}(A) = k$, every subset of $k$ columns of $A$ is linearly independent. Motivated by this property, we introduce the following penalty term that promotes the K-rank of a given matrix $A$:

$$\mathcal{R}_{\text{K}-\text{rank}}(A) = - \sum_{J_k \in \binom{[c_{in}]}{k}} \log \det(A_{:,J_k}^\top A_{:,J_k} + \epsilon I),$$
$$(6)$$

where $J_k \in \binom{[c_{in}]}{k}$ is selected from the set of all subsets of size $k$ drawn from the index set $[c_{in}] = \{1, 2, \cdots, c_{in}\}$, $\det(\cdot)$ denotes the determinant of a matrix, $I$ is the identity matrix and $\epsilon = 1e - 6$ is used to prevent the numerical error. In the following, we denote $A_k \triangleq A_{:,J_k}$ for simplicity.

Equation (6) is designed to encourage linear independence among the columns of a sub-matrix $A_k \in \mathbb{R}^{r \times k}$. Specifically, $A_k^\top A_k$ is the Gram matrix of the $k$ column vectors, and its determinant admits a geometric interpretation: $\det(A_k^\top A_k)$ equals the squared $k$-dimensional volume of the parallel polyhedron spanned by these columns. When the columns are linearly independent, $A_k$ has full column rank, hence $A_k^\top A_k$ is positive definite and $\det(A_k^\top A_k) > 0$; when the columns become linearly dependent, $A_k^\top A_k$ becomes singular and its determinant collapses to zero. Therefore, maximizing $\det(A_k^\top A_k)$ (equivalently, maximizing $\log \det(A_k^\top A_k)$) promotes larger spanned volume and thus stronger linear independence. To improve numerical stability, we use $\log \det(A_k^\top A_k + \varepsilon I)$ with a small $\varepsilon > 0$, without changing the overall objective of encouraging column independence. Finally, Equation (6) aggregates this criterion over all $k$-column sub-matrices, yielding a penalty term that encourages a higher K-rank for $A$.

In practice, we found that calculating Equation (6) at every LoRA training step is computationally expensive. Thus,

we make the following two simplifications to Equation (6). Firstly, instead of calculating the log determinant of *all* submatrices with $k$ columns, we only compute it for *one* randomly selected sub-matrix at each training step. Secondly, instead of selecting $k$ columns from the original matrix, we randomly sample an index $k' \sim \text{Uniform}\{2, \cdots, k\}$ and select $k'$ columns from the original matrix to optimize. The simplified penalty term is defined as:

$$\mathcal{R}'_{\text{K}-\text{rank}}(A) = - \log \det(A_{:,J_{k'}}^\top A_{:,J_{k'}} + \epsilon I),$$
$$s.t. \quad k' \sim \text{Uniform}\{2, \cdots, k\}. \quad (7)$$

We set $k = r$ during the experiment since the $k$-rank will not exceed $r$ given $A \in \mathbb{R}^{r \times c_{in}}$. Note that the second simplification is reasonable since we aim to maximize the K-rank of the matrix $A$. When the penalty term is incorporated into the task loss, the resulting K-rank may fall below $r$ to maintain the quality of task optimization. Thus, the second modification enables maximizing the K-rank within a narrower range of values.

The final loss function is computed as:

$$\mathcal{L}_{\text{total}} = \mathcal{L}_{\text{task}} + \lambda \cdot \sum_{\ell=1}^{L} \left[ \mathcal{R}'_{\text{K}-\text{rank}}(A^\ell) + \mathcal{R}'_{\text{K}-\text{rank}}(B^{\ell\top}) \right],$$
$$(8)$$

where $L$ is the total number of matrices that need to be fine-tuned via LoRA, $\lambda$ is the hyper-parameter to control the influence of the penalty term, and $\mathcal{L}_{\text{task}}$ is the task loss determined by the downstream tasks. The detailed algorithm for our proposed LoKRA is shown in Appendix D.

### 4.3. LoKRA$^+$

The analysis above is derived from the upper bound in Theorem 4.1. Unfortunately, even when $\Delta W$ contains no all-zero columns, the LoRA-style matrix product $\Delta W = BA$ still lacks a meaningful lower-bound guarantee on its K-rank under our regularization. Although this weak lower bound is rarely approached during training and thus typically has little influence on empirical results, it reveals a theoretical mismatch: plain matrix multiplication is not inherently well aligned with promoting K-rank.

In this subsection, we aim to replace the matrix multiplication in $\Delta W = BA$ with an alternative operation, the Khatri-Rao product, which is theoretically better suited to K-rank.

**Definition 4.2** (Khatri-Rao product)**.** Given two matrices $C \in \mathbb{R}^{q \times c_{in}}$ and $D \in \mathbb{R}^{p \times c_{in}}$, the Khatri-Rao product of $C$ and $D$ is defined as:

$$D \odot C = [d_1 \otimes c_1, d_2 \otimes c_2, \cdots, d_{c_{in}} \otimes c_{c_{in}}] \in \mathbb{R}^{pq \times c_{in}}, \quad (9)$$

where $\odot$ stands for Khatri-Rao product, $c_i$ ($d_i$) is the $i$-th column of $C$ ($D$), and $d_i \otimes c_i$ is the Kronecker product

which is defined as:

$$
\begin{aligned}
\boldsymbol{d}_i \otimes \boldsymbol{c}_i &= [d_{1i}\boldsymbol{c}_i, d_{2i}\boldsymbol{c}_i, \cdots, d_{pi}\boldsymbol{c}_i]^\top \\
&= [d_{1i}c_{1i}, \cdots, d_{1i}c_{qi}, \cdots, d_{pi}c_{1i}, \cdots, d_{pi}c_{qi}]^\top \\
&\in \mathbb{R}^{pq}.
\end{aligned}
\tag{10}
$$

The forward pass of the standard LoRA in Equation (1) is replaced by:

$$
f(\boldsymbol{x}) = (\boldsymbol{W} + \Delta\boldsymbol{W}')\boldsymbol{x} = \boldsymbol{W}\boldsymbol{x} + (\boldsymbol{D} \odot \boldsymbol{C})\boldsymbol{x}. \tag{11}
$$

**Analysis of K-rank.** Assuming that no column in $\Delta\boldsymbol{W}'$ is all-zero column, the K-rank of $\Delta\boldsymbol{W}' = \boldsymbol{D} \odot \boldsymbol{C}$ is bounded by the following theorem. The proof is shown in Appendix B.

**Theorem 4.3.** *Given two matrices $\boldsymbol{C} \in \mathbb{R}^{q \times c_{in}}$ and $\boldsymbol{D} \in \mathbb{R}^{p \times c_{in}}$, the K-rank of the Khatri-Rao product result is bounded by:*

$$
\min\{\mathrm{kr}(\boldsymbol{C}) + \mathrm{kr}(\boldsymbol{D}) - 1, c_{in}\} \leq \mathrm{kr}(\boldsymbol{D} \odot \boldsymbol{C}) \leq c_{in}. \tag{12}
$$

Compared to column K-rank, the row K-rank of the Khatri-Rao product does not have a specific relationship between the row K-rank of $\boldsymbol{D}$ and $\boldsymbol{C}$, as shown in the theorem below (see the example in Appendix C):

**Theorem 4.4.** *Given two matrices $\boldsymbol{C} \in \mathbb{R}^{q \times c_{in}}$ and $\boldsymbol{D} \in \mathbb{R}^{p \times c_{in}}$, the row K-rank of the Khatri-Rao product result is bounded by:*

$$
1 \leq \mathrm{kr}_{\mathrm{row}}(\boldsymbol{D} \odot \boldsymbol{C}) \leq c_{out}. \tag{13}
$$

This is because each row of the Khatri-Rao product is the Hadamard product of the corresponding rows of $\boldsymbol{D}$ and $\boldsymbol{C}$, and generally introduces additional zero entries, thereby increasing sparsity in the resulting matrix and leading to an unbounded row K-rank. Thus, we only concentrate on improving the column K-rank of $\boldsymbol{D}$ and $\boldsymbol{C}$ in the following.

Similar to Equation (8), the final loss function of LoKRA$^+$ is computed as:

$$
\mathcal{L}_{\mathrm{total+}} = \mathcal{L}_{\mathrm{task}} + \lambda \cdot \sum_{\ell=1}^{L} \left[ \mathcal{R}'_{\mathrm{K-rank}}(\boldsymbol{C}^\ell) + \mathcal{R}'_{\mathrm{K-rank}}(\boldsymbol{D}^\ell) \right],
\tag{14}
$$

In LoKRA$^+$, we have $\Delta\boldsymbol{W}' = \boldsymbol{D} \odot \boldsymbol{C}$ where $\Delta\boldsymbol{W}' \in \mathbb{R}^{c_{out} \times c_{in}}$, $\boldsymbol{C} \in \mathbb{R}^{q \times c_{in}}$ and $\boldsymbol{D} \in \mathbb{R}^{p \times c_{in}}$. Therefore, $pq = c_{out}$ is an implicit default condition. There are multiple pairs $(p, q)$ that satisfy this condition; we propose to choose the pair that has the minimum trainable parameters:

$$
(p^*, q^*) = \arg\min_{pq=c_{out}} |p - q|. \tag{15}
$$

Given that the product of $p$ and $q$ is fixed, the network has the fewest parameters when their difference $|p - q|$ is

minimized. In the following experiments, we use $(p^*, q^*)$ with $p^* \leq q^*$ by default. Similarly to (Hu et al., 2022), we initialize $\boldsymbol{C}$ with random Gaussian values (He et al., 2015) and $\boldsymbol{D}$ with zeros. Based on the consistency of the adapters and $\delta\boldsymbol{W}$ shapes, LoKRA$^+$ does not affect the inference structure and the merge process. Limitations of our methods are discussed in Appendix G.

# 5. Experiments

In this section, we conduct a series of experiments to validate the effectiveness of the proposed LoKRA and LoKRA$^+$. Our methods are first compared with several PEFT methods on the commonsense reasoning task. Then, we extend our method to math-reasoning scenarios and demonstrate that the gains remain consistent from supervised fine-tuning (SFT) to reinforcement learning. Finally, several ablation studies are performed to evaluate the impact of our methods.

## 5.1. Experimental Settings

**Models**. We adapt pre-trained LLMs to downstream tasks with LoKRA, LokRA$^+$, and other PEFT methods. For commonsense reasoning, we apply supervised fine-tuning (SFT) to LLaMA-7B/13B (Touvron et al., 2023a), LLaMA2-7B (Touvron et al., 2023b), LLaMA3-8B (Dubey et al., 2024), and Qwen3-8B (Yang et al., 2025b). For math reasoning, we perform RL fine-tuning on DeepSeek-R1-Distill-Qwen-1.5B (Guo et al., 2025).

**Tasks and benchmarks**. We evaluate our method on downstream adaptation for both commonsense reasoning and math reasoning tasks. For commonsense reasoning, we follow the standard commonsense_reasoning_170k protocol: we merge the training splits of BoolQ (Clark et al., 2019), PIQA (Bisk et al., 2020), Social IQa (Sap et al., 2019), HellaSwag (Zellers et al., 2019), WinoGrande (Sakaguchi et al., 2021), ARC-Easy/Challenge (Clark et al., 2018), and OpenBookQA (Mihaylov et al., 2018), then sub-sample 170,420 instruction-response pairs for supervised fine-tuning. We report results on the official test sets of all previous benchmarks.

For math reasoning, we use GRPO (Guo et al., 2025) as the reinforcement learning algorithm to train the mathematical reasoning ability of the base model DeepSeek-R1-Distill-Qwen-1.5B (Guo et al., 2025). We consider five training pipelines built on DeepSeek-R1-Distill-Qwen-1.5B, each with a different data recipe and resulting model. (1) STILL-3-1.5B-preview (RUCAIBox STILL Team, 2025) is trained on MATH (Hendrycks et al., 2021; Lightman et al., 2023), NuminaMathCoT (Li et al., 2024), and AIME (1983–2023) (Art of Problem Solving, 2024). (2) DeepScaleR-1.5B-Preview (Luo et al., 2025) is trained on over 40k samples from AIME (Art of Problem Solving, 2024), AMC (Art

of Problem Solving, 2023), OMNI-MATH (Gao et al., 2024), and STILL (RUCAIBox STILL Team, 2025). (3) Open-RS1/2/3 (Dang & Ngo, 2025) are three pipelines trained with the s1 (Muennighoff et al., 2025) and DeepScaleR (Luo et al., 2025) datasets. We evaluate all resulting models on MATH500 (Hendrycks et al., 2021; Lightman et al., 2023), AIME (2024–2025) (Art of Problem Solving, 2024), AMC2023 (Art of Problem Solving, 2023), GPQA (Rein et al., 2024), and Minerva (Nagrani et al., 2025).

## 5.2. Performance on Commonsense Reasoning

**Baselines**. We compare LoKRA and LoKRA$^+$ with a bunch of PEFT methods including LoRA-based methods such as LoRA (Hu et al., 2022), DoRA (Liu et al., 2024), RandLoRA (Albert et al.), MoRA (Jiang et al., 2024), NoRA+ (Li et al., 2025), SHiRA-SNIP (Bhardwaj et al., 2024), MiLoRA (Wang et al., 2025a) and HiRA (Huang et al., 2025), and prompt-based methods such as parallel adapter tuning (Adapter-P) (He et al., 2021) and P-tuning (Liu et al., 2022). Experimental settings are the same as DoRA, except that the K-rank loss is added to the original task loss.

**Results.** As shown in Table 1, LoKRA consistently outperforms LoRA by optimizing the K-rank of the update matrix, which better alleviates redundancy in parameter updates. Under the same trainable-parameter budget, our objective in Equation (8) improves average accuracy by **2.4%-5.0%** across all base models, and surpasses the previous best method by 0.3%, 0.5%, 0.6%, 0.3%, and 1.5% on LLaMA-7B, LLaMA-13B, LLaMA2-7B, LLaMA3-8B, and Qwen3-8B, respectively. LoKRA$^+$ further improves over LoKRA by 0.3%, 0.8%, 0.6%, 0.4%, and 0.3%. Although LoKRA$^+$ introduces additional trainable parameters, it still achieves the best overall performance and outperforms LoRA under the same parameter budget (see Figure 1 for details). Finally, since HiRA reports results under a different training-evaluation protocol (*e.g.*, best-checkpoint selection and a different HuggingFace trainer), we re-implement HiRA under the same setting as DoRA for a fair comparison. Results under the original HiRA protocol are provided in Appendix E.

## 5.3. Performance on Math Reasoning

**Baselines**. We conduct experiments and compare LoKRA and LoKRA$^+$ with Tina (Wang et al., 2025b), a TRL-based (von Werra et al., 2020) framework that incorporates LoRA into the RL training process and generates a series of reasoning models using DeepSeek-R1-Distill-Qwen-1.5B as the base model. The full-parameter fine-tuning is also compared. Our methods have exactly the same experimental setting as Tina, except that the K-rank loss is added to the original task loss.

**Results.** As shown in Table 2, we conduct experiments

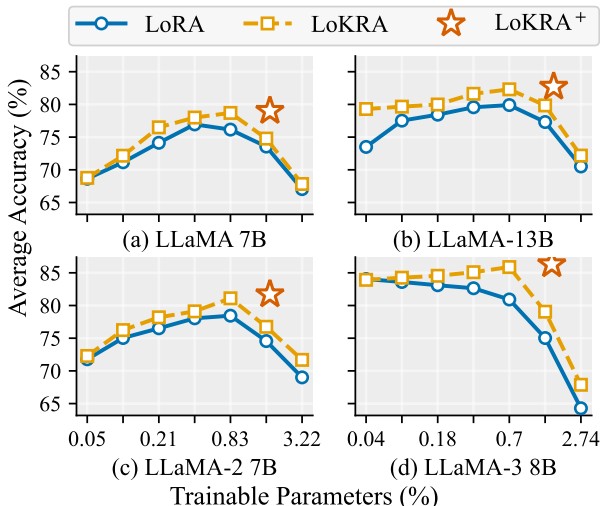

*Figure 1.* Average accuracies of using different trainable parameters for LoRA and LoKRA. LoKRA$^+$ uses a fixed number of trainable parameters. The experiments are conducted on commonsense reasoning datasets with LLaMA-7B/13B, LLaMA2-7B and LLaMA3-8B.

on five reasoning models and compare the average accuracy result over six math reasoning datasets. Among all the results, the proposed LoKRA and LoKRA$^+$ outperform Tina and the baseline by a large margin. Specifically, LoKRA surpasses Tina with LoRA-based RL strategy by 3.75%, 2.87%, 1.58%, 0.35%, and 1.28% on reasoning models STILL-3-1.5B-preview, DeepScalerR-1.5B-preview, and Open-RS1/RS2/RS3. LoKRA$^+$ further improves the performance of LoKRA by 0.34%, 1.15%, 1.32%, 0.53%, and 1.05%. These results show that our proposed methods are effective not only under supervised fine-tuning but also for reasoning models trained with reinforcement learning.

## 5.4. Ablation Studies

We conduct several ablation studies to further validate the effectiveness of our method.

**Different trainable parameters.** In Table 1, the proposed LoKRA$^+$ has more trainable parameters compared to other methods, and raises an unfair comparison risk. Thus, in Figure 1 we extend LoRA and LoKRA to different ranks and compare them to LoKRA$^+$. In our experiments, we find that increasing the number of trainable parameters leads to an initial rise, followed by a drop in LoRA's average accuracy, a trend that aligns with conclusions from previous studies (Hu et al., 2022; Liu et al., 2024). The proposed LoKRA consistently outperforms LoRA across different parameter counts and baseline models, while using the same number of trainable parameters. Although LoKRA$^+$ has a relatively fixed number of trainable parameters, it achieves

*Table 1.* The results of the proposed LoKRA and LoKRA$^+$, and other baselines with LLaMA-7B/13B, LLaMA2-7B, LLaMA3-8B, and Qwen3-8B on commonsense reasoning datasets. For all metrics, higher is better. The best performance is **bolded**, and the second best is highlighted in underline.

| Model | Methods | Trainable Param. (%) | Commonsense Reasoning Datasets | | | | | | | | |
|---|---|---|---|---|---|---|---|---|---|---|---|
| | | | BoolQ | PIQA | SIQA | HellaS | WinoG | ARC-e | ARC-c | OBQA | Avg. |
| L-7B | Adapter-P | 3.54 | 67.9 | 76.4 | 78.8 | 69.8 | 78.9 | 73.7 | 57.3 | 75.2 | 72.2 |
| | HiRA ($r = 32$) | 0.83 | 70.0 | 82.4 | 79.2 | 85.1 | 80.4 | 82.1 | 64.8 | 77.8 | 77.7 |
| | LoRA ($r = 32$) | 0.83 | 67.5 | 80.8 | 78.2 | 83.4 | 80.4 | 78.0 | 62.6 | 79.1 | 76.3 |
| | LoRA ($r = 64$) | 1.64 | 66.7 | 79.1 | 75.7 | 17.6 | 78.8 | 73.3 | 59.6 | 75.2 | 65.8 |
| | DoRA ($r = 32$) | 0.84 | 69.7 | **83.4** | 78.6 | 87.2 | **81.0** | 81.9 | 66.2 | 79.2 | 78.4 |
| | DoRA ($r = 64$) | 1.65 | 70.1 | 82.0 | 75.6 | 85.9 | 79.7 | 79.1 | 63.7 | 78.4 | 76.8 |
| | NoRA+ ($r = 32$) | 0.83 | 69.9 | 81.8 | 77.4 | 82.1 | 80.0 | 79.7 | 64.3 | 78.6 | 76.7 |
| | SHiRA-SNIP | 1.00 | 68.3 | 80.6 | 79.1 | 82.1 | 80.0 | 81.5 | 67.9 | 79.6 | 77.4 |
| | LoKRA ($r = 32$) | 0.83 | 69.8 | 82.2 | 78.1 | **87.5** | 80.5 | 82.3 | 67.2 | **82.0** | 78.7 |
| | LoKRA$^+$ | 1.80 | **70.5** | 83.0 | **79.5** | 86.0 | 80.5 | **82.9** | **68.5** | 81.0 | **79.0** |
| L-13B | HiRA ($r = 32$) | 0.67 | 72.8 | 85.7 | 79.5 | 91.5 | 83.7 | 85.6 | 71.4 | 84.2 | 81.8 |
| | LoRA ($r = 32$) | 0.67 | 71.6 | 83.4 | 80.0 | 89.9 | 84.2 | 81.2 | 67.7 | 80.8 | 79.9 |
| | LoRA ($r = 64$) | 1.33 | 69.7 | 82.0 | 79.0 | 86.2 | 81.5 | 75.3 | 65.5 | 79.4 | 77.3 |
| | DoRA ($r = 32$) | 0.68 | 72.4 | 84.9 | **81.2** | 91.5 | 83.7 | 84.6 | 68.9 | 81.6 | 81.1 |
| | DoRA ($r = 64$) | 1.34 | 71.8 | 82.4 | 81.1 | 89.2 | **85.8** | 83.4 | 71.1 | 82.0 | 80.8 |
| | LoKRA ($r = 32$) | 0.67 | 73.7 | 85.8 | 80.1 | 92.7 | 84.5 | 84.9 | **73.7** | 83.0 | 82.3 |
| | LoKRA$^+$ | 1.64 | **73.8** | **86.5** | 81.0 | **93.3** | 84.9 | **85.9** | 72.6 | **86.4** | **83.1** |
| L2-7B | P-Tuning | 0.74 | 58.8 | 36.0 | 0.2 | 0.0 | 0.0 | 2.0 | 0.2 | 0.8 | 12.2 |
| | MoRA ($r = 32$) | 0.82 | 72.2 | 80.8 | 79.5 | 29.1 | 80.2 | 85.3 | 71.4 | 81.2 | 72.5 |
| | HiRA ($r = 32$) | 0.83 | 71.9 | 83.9 | 78.6 | **91.2** | 81.8 | 84.5 | 69.4 | 82.4 | 80.5 |
| | LoRA ($r = 32$) | 0.83 | 68.9 | 82.2 | 78.1 | 86.9 | 81.2 | 79.3 | 65.4 | 78.4 | 77.6 |
| | LoRA ($r = 64$) | 1.64 | 69.9 | 82.4 | 78.0 | 87.9 | 82.2 | 81.0 | 67.5 | 78.6 | 78.4 |
| | DoRA ($r = 32$) | 0.84 | 71.8 | 83.7 | 76.0 | 89.1 | 82.6 | 83.7 | 68.2 | 82.4 | 79.7 |
| | DoRA ($r = 64$) | 1.65 | 66.5 | 81.6 | 80.0 | 84.1 | 81.8 | 82.4 | 69.4 | 80.2 | 78.2 |
| | NoRA+ ($r = 32$) | 0.83 | 70.5 | 81.9 | 79.1 | 87.7 | 82.2 | 82.7 | 67.1 | 80.2 | 78.9 |
| | SHiRA-SNIP | 1.00 | 70.4 | 81.7 | 79.0 | 89.8 | 80.5 | 83.3 | 68.6 | 81.0 | 79.3 |
| | MiLoRA ($r = 32$) | 0.83 | 67.6 | 83.8 | 80.1 | 88.2 | 82.0 | 82.8 | 68.8 | 80.6 | 79.2 |
| | LoKRA ($r = 32$) | 0.83 | **72.8** | 83.8 | **80.3** | 89.5 | 82.8 | 85.0 | 72.1 | 82.4 | 81.1 |
| | LoKRA$^+$ | 1.80 | 72.5 | **84.6** | 79.5 | 90.6 | **84.1** | **85.7** | **73.0** | **83.4** | **81.7** |
| L3-8B | P-Tuning | 0.62 | 60.0 | 11.6 | 8.2 | 1.8 | 37.7 | 8.6 | 7.4 | 9.6 | 18.1 |
| | RandLoRA | 0.70 | **76.3** | 88.1 | 80.3 | **95.7** | 86.1 | 90.4 | 80.9 | 87.0 | 85.6 |
| | MoRA ($r = 32$) | 0.70 | 74.3 | 87.4 | 80.7 | 43.5 | 86.7 | 91.2 | 79.6 | 85.6 | 78.6 |
| | HiRA ($r = 32$) | 0.70 | 75.4 | 88.2 | 80.7 | 95.6 | 85.5 | 91.0 | 80.0 | 86.8 | 85.4 |
| | LoRA ($r = 32$) | 0.70 | 71.2 | 85.1 | 79.3 | 92.1 | 82.6 | 85.2 | 70.1 | 81.4 | 80.9 |
| | LoRA ($r = 64$) | 1.39 | 67.9 | 80.7 | 77.0 | 83.4 | 78.1 | 76.6 | 60.9 | 76.4 | 75.1 |
| | DoRA ($r = 32$) | 0.71 | 74.6 | **89.3** | 79.9 | 95.5 | 85.6 | 90.5 | 80.4 | 85.8 | 85.2 |
| | DoRA ($r = 64$) | 1.40 | 73.3 | 88.4 | 80.2 | 94.3 | 85.4 | 89.4 | 78.9 | 87.0 | 84.6 |
| | MiLoRA ($r = 32$) | 0.70 | 68.8 | 86.7 | 77.2 | 92.9 | 85.6 | 86.8 | 75.5 | 81.8 | 81.9 |
| | LoKRA ($r = 32$) | 0.70 | 74.9 | 89.1 | **81.3** | 95.5 | **86.9** | 90.5 | 81.2 | 87.6 | 85.9 |
| | LoKRA$^+$ | 1.52 | 76.0 | 88.5 | 81.1 | **95.7** | 86.0 | **91.9** | 81.6 | **89.6** | **86.3** |
| Q3-8B | LoRA (r=32) | 0.71 | 73.4 | 89.2 | 80.1 | 83.4 | 85.4 | 96.0 | 87.8 | 89.6 | 85.6 |
| | LoRA (r=64) | 1.42 | 71.4 | 87.9 | 79.5 | 93.6 | 83.8 | 94.6 | 86.3 | 90.8 | 86.0 |
| | DoRA (r=32) | 0.72 | 73.3 | 89.3 | 80.5 | 94.3 | 85.6 | 95.7 | 88.8 | 91.4 | 87.4 |
| | DoRA (r=64) | 1.43 | 73.7 | 89.3 | 80.3 | 94.5 | 86.7 | 95.9 | 89.2 | 92.60 | 87.8 |
| | LoKRA ($r = 32$) | 0.71 | 74.2 | 90.5 | 81.6 | 95.3 | **86.9** | **97.2** | **92.6** | **92.8** | 88.9 |
| | LoKRA$^+$ | 1.53 | **75.9** | **90.9** | **82.1** | **95.8** | 86.7 | **97.2** | 91.8 | **92.8** | **89.2** |

the best performance among all the baseline models.

**Align the training time.** There are additional calculations in the proposed loss function Equation (8) and Equation (14).

Therefore, we plot the validation loss curve of different methods on LLaMA-7B using the same amount of training time in Figure 2, and also report the corresponding average

*Table 2.* The results of the proposed LoKRA, LoKRA$^+$ and other baselines with different models on six math reasoning datasets. For all metrics, higher is better.

| Pipeline | Methods | AIME24 | AIME25 | AMC23 | MATH500 | GPQA | Minerva | Avg. |
|---|---|---|---|---|---|---|---|---|
| STILL-3-1.5B-preview | FFT | 26.67 | 26.67 | 67.50 | 86.40 | 34.34 | 27.57 | 44.86 |
| | Tina | 36.67 | 30.00 | 77.50 | 84.60 | 33.33 | 26.84 | 48.16 |
| | LoKRA | 50.00 | 26.67 | 85.00 | 85.00 | 32.83 | 31.99 | 51.91 |
| | LoKRA$^+$ | 43.33 | 30.00 | 87.50 | 84.00 | 38.89 | 29.78 | 52.25 |
| DeepScalerR-1.5B-preview | FFT | 36.67 | 26.67 | 77.50 | 87.80 | 31.82 | 31.99 | 48.74 |
| | Tina | 43.33 | 26.67 | 67.50 | 86.20 | 37.88 | 28.68 | 48.38 |
| | LoKRA | 36.67 | 26.67 | 82.50 | 83.80 | 46.97 | 30.88 | 51.25 |
| | LoKRA$^+$ | 50.00 | 30.00 | 80.00 | 85.20 | 39.39 | 29.78 | 52.40 |
| Open-RS1 | FFT | 26.67 | 20.00 | 72.50 | 83.60 | 35.35 | 28.68 | 44.47 |
| | Tina | 43.33 | 20.00 | 80.00 | 84.00 | 35.35 | 28.68 | 48.56 |
| | LoKRA | 40.00 | 30.00 | 80.00 | 85.40 | 33.84 | 31.62 | 50.14 |
| | LoKRA$^+$ | 43.33 | 26.67 | 82.50 | 85.40 | 38.89 | 31.99 | 51.46 |
| Open-RS2 | FFT | 26.67 | 13.33 | 62.50 | 85.40 | 34.85 | 26.84 | 41.60 |
| | Tina | 43.33 | 26.67 | 77.50 | 87.00 | 36.36 | 32.72 | 50.60 |
| | LoKRA | 50.00 | 26.67 | 72.50 | 84.40 | 40.91 | 31.25 | 50.95 |
| | LoKRA$^+$ | 36.67 | 33.33 | 90.00 | 86.20 | 34.34 | 28.31 | 51.48 |
| Open-RS3 | FFT | 43.33 | 20.00 | 67.50 | 83.00 | 33.84 | 28.68 | 46.06 |
| | Tina | 36.67 | 23.33 | 82.50 | 85.20 | 37.37 | 31.62 | 49.45 |
| | LoKRA | 43.33 | 30.00 | 75.00 | 84.00 | 40.40 | 31.62 | 50.73 |
| | LoKRA$^+$ | 40.00 | 30.00 | 87.50 | 84.40 | 39.39 | 29.41 | 51.78 |

*Table 3.* The average accuracies of LoRA, LoKRA, and LoKRA$^+$ with LLaMA-7B on commonsense reasoning datasets using the same amount of training time as LoKRA$^+$.

| Method | Average Accuracy |
|---|---|
| LoRA | 76.3→72.7 |
| LoKRA | 78.7→77.8 |
| LoKRA$^+$ | 79.0 |

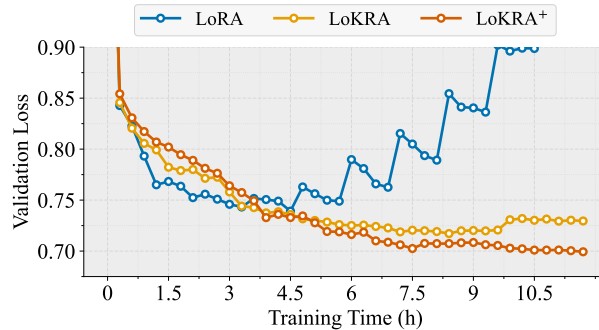

*Figure 2.* The validation loss curve of LoRA, LoKRA and LoKRA$^+$ using the same amount of training time as LoKRA$^+$.

accuracies on commonsense reasoning datasets in Table 3. The results show that LoRA and LoKRA do not benefit from additional training steps, as overfitting may occur under these conditions. Specifically, by using the same amount of training time as LoKRA$^+$ does, the average accuracy of LoRA drops from 76.3% to 72.7%, and LoKRA drops from 78.7% to 77.8%. Therefore, this experiment shows that although LoKRA$^+$ requires more computational resources and time, it can achieve a better performance.

**Different loss functions.** In Section 4.2, we simplified the original K-rank loss Equation (6) into Equation (7) to avoid spending too much time on training. Specifically, we use 1) $k' \sim \text{Uniform}\{2, \cdots k\}$ to replace $k$, and 2) calculation on one randomly selected sub-matrix instead of all sub-matrices. Therefore, in Table 4 we remove the first sim-

plification and study the relative training time and inference accuracy of LLaMA-7B. By using $k$ instead of $k'$, the training time increases by $1.1\times$, and the average accuracy shows no obvious improvement. The second simplification cannot be removed, otherwise we need to compute the determinant of $C_{4096}^{32} \approx 10^{80}$ sub-matrices for each layer in LLaMA-7B, which is unrealistic.

More ablation studies are shown in Appendix F.

*Table 4.* The average accuracy and relative training time of removing simplifications in Equation (7). Experiments are conducted on LLaMA-7B model using commonsense reasoning datasets.

| Method | Relative Train. | Acc. |
|---|---|---|
| LoKRA | 1.0× | 78.7 |
| LoKRA w/o simplification | 1.1× | 78.8 |

# 6. Conclusion

In this paper, we revisit the rank of LoRA updates and show that the conventional matrix rank is insufficient to identify redundancy and duplicated directions in parameter updates. To solve this problem, we propose a new PEFT algorithm called LoKRA that theoretically guarantees a higher Kruskal rank for the update matrix and minimizes redundancy under limited trainable parameters, by maximizing the log-determinant of each sub-matrix to promote column-wise linear independence. A LoKRA$^+$ algorithm that replaces matrix multiplication with Khatri-Rao product is further studied to better align with Kruskal-rank. Experimental results demonstrate the superiority of our methods.

# Impact Statement

This paper presents work whose goal is to advance the field of large language models. There are many potential societal consequences of our work, none of which we feel must be specifically highlighted here.

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

## A. Proof of Theorem 4.1

We first give the proof of the K-rank (Inequation 4).

By definition, the K-rank of a given matrix is always less than or equal to the rank of the same matrix. Since $\mathrm{rank}(\boldsymbol{B}\boldsymbol{A}) \le r$, we have $\mathrm{kr}(\boldsymbol{B}\boldsymbol{A}) \le r$. Given that no column in $\Delta\boldsymbol{W}$ is an all-zero column, it is straightforward that $\mathrm{kr}(\boldsymbol{B}\boldsymbol{A}) \ge 1$.

Suppose $\mathrm{kr}(\boldsymbol{A}) = k_A$, then there exist $k_A + 1$ columns in $\boldsymbol{A}$ that are linearly dependent. Define the sub-matrix of these $k_A + 1$ columns as $\boldsymbol{A}_{:,\mathrm{sub}}$. Then, there exist a non-zero vector $\boldsymbol{d} \in \mathbb{R}^{k_A+1}$ such that

$$\boldsymbol{A}_{:,\mathrm{sub}}\boldsymbol{d} = \boldsymbol{0}. \tag{16}$$

Accordingly, since $\boldsymbol{A}_{:,\mathrm{sub}}\boldsymbol{d} = \boldsymbol{0}$, left-multiplying by $\boldsymbol{B}$ gives

$$\boldsymbol{B}\,\boldsymbol{A}_{:,\mathrm{sub}}\boldsymbol{d} = \boldsymbol{B}\big(\boldsymbol{A}_{:,\mathrm{sub}}\boldsymbol{d}\big) = \boldsymbol{0}, \tag{17}$$

which means that $\mathrm{kr}(\boldsymbol{B}\boldsymbol{A}) \le \mathrm{kr}(\boldsymbol{A})$.

Also, given that the rank of a matrix is always no less than the K-rank of a matrix, and that $\mathrm{rank}(\boldsymbol{B}\boldsymbol{A}) \le \mathrm{rank}(\boldsymbol{B})$, we have:

$$\mathrm{kr}(\boldsymbol{B}\boldsymbol{A}) \le \mathrm{rank}(\boldsymbol{B}\boldsymbol{A}) \le \mathrm{rank}(\boldsymbol{B}). \tag{18}$$

Thus, we finish the proof of Inequation 4.

Then, we give the proof of Inequation 5. It is easy to know that

$$\mathrm{kr}_{\mathrm{row}}(\boldsymbol{B}\boldsymbol{A}) = \mathrm{kr}((\boldsymbol{B}\boldsymbol{A})^\top) = \mathrm{kr}(\boldsymbol{A}^\top\boldsymbol{B}^\top). \tag{19}$$

By reusing the conclusion in Inequation 4, we finish the proof of Inequation 5. Therefore, we finish the proof of Theorem 4.1.

## B. Proof of Theorem 4.3

The proof originates from (Ten Berge, 2000). It is obvious that the K-rank of a matrix should be no more than the number of columns of that matrix. Thus, we have

$$\mathrm{kr}(\boldsymbol{D} \odot \boldsymbol{C}) \le c_{in}. \tag{20}$$

Define $S$ as the smallest number of linearly dependent columns of $\boldsymbol{D} \odot \boldsymbol{C}$. Then $\mathrm{kr}(\boldsymbol{D} \odot \boldsymbol{C}) = S - 1$. Let $\boldsymbol{D}_S \odot \boldsymbol{C}_S$ collect such an $S$-column dependent subset, where $\boldsymbol{D}_S$ and $\boldsymbol{C}_S$ contain the corresponding columns of $\boldsymbol{D}$ and $\boldsymbol{C}$, respectively. Hence there exists a nonzero vector $\boldsymbol{v}_S \in \mathbb{R}^S$ such that $(\boldsymbol{D}_S \odot \boldsymbol{C}_S)\boldsymbol{v}_S = \boldsymbol{0}$, which means $\boldsymbol{D}_S \boldsymbol{V}_S \boldsymbol{C}_S^\top = \boldsymbol{0}$, where $\boldsymbol{V}_S = \mathrm{diag}(\boldsymbol{v}_S)$. Applying Sylvester's inequality to $\boldsymbol{D}_S(\boldsymbol{V}_S\boldsymbol{C}_S^\top)$ gives:

$$0 = \mathrm{rank}(\boldsymbol{D}_S\boldsymbol{V}_S\boldsymbol{C}_S^\top) \ge \mathrm{rank}(\boldsymbol{D}_S) + \mathrm{rank}(\boldsymbol{V}_S\boldsymbol{C}_S^\top) - S. \tag{21}$$

Since $\boldsymbol{V}_S$ is nonsingular, $\mathrm{rank}(\boldsymbol{V}_S\boldsymbol{C}_S^\top) = \mathrm{rank}(\boldsymbol{C}_S^\top) = \mathrm{rank}(\boldsymbol{C}_S)$. Thus,

$$\mathrm{kr}(\boldsymbol{D} \odot \boldsymbol{C}) \ge \mathrm{rank}(\boldsymbol{D}_S) + \mathrm{rank}(\boldsymbol{C}_S) - 1. \tag{22}$$

Since $\mathrm{rank}(\boldsymbol{D}_S) \ge \mathrm{kr}(\boldsymbol{D}_S) \ge \mathrm{kr}(\boldsymbol{D})$ and $\mathrm{rank}(\boldsymbol{C}_S) \ge \mathrm{kr}(\boldsymbol{C}_S) \ge \mathrm{kr}(\boldsymbol{C})$, we complete the proof.

## C. Example of Khatri-Rao Product Achieving the Lower-Bound in Theorem 4.4

In this section, we give an example of the Khatri-Rao product: the inputs have row K-rank greater than 1, but the output has row K-rank 1.

Given

$$\boldsymbol{D} = \begin{pmatrix} 1 & 1 & 1 \\ 0 & 1 & 2 \end{pmatrix}, \qquad \boldsymbol{C} = \begin{pmatrix} 1 & 1 & 1 \\ 0 & 1 & 2 \end{pmatrix},$$

we can compute the Khatri-Rao product of $\boldsymbol{D}$ and $\boldsymbol{C}$ as:

$$\boldsymbol{D} \odot \boldsymbol{C} = \begin{pmatrix} 1 & 1 & 1 \\ 0 & 1 & 2 \\ 0 & 1 & 2 \\ 0 & 1 & 4 \end{pmatrix}.$$

It is easy to know that $\mathrm{kr}_{\mathrm{row}}(\boldsymbol{D}) = 2$, $\mathrm{kr}_{\mathrm{row}}(\boldsymbol{C}) = 2$, and $\mathrm{kr}_{\mathrm{row}}(\boldsymbol{D} \odot \boldsymbol{C}) = 1$.

## D. Algorithm of LoKRA

In this section, we detail the algorithm of our proposed LoKRA.

---

**Algorithm 1** Low Kruskal-Rank Adaptation (LoKRA)

---

**Input:** Pretrained weight matrices $\{\boldsymbol{W}^\ell\}_{\ell=1}^L$, penalty term hyper-parameter $\lambda$, training data $\mathcal{D}$, learning rate $\eta$, total training epochs $T$, total number of matrices $L$ to be fine-tuned in the network

**Output:** Adapted weight matrices $\{\boldsymbol{W}_{out}^\ell \leftarrow \boldsymbol{W}^\ell + \Delta\boldsymbol{W}^\ell\}_{\ell=1}^L$

1: Kaiming initialize $\{\boldsymbol{A}^\ell \in \mathbb{R}^{r \times c_{in}^\ell}\}_{\ell=1}^L$ with random Gaussian values
2: Zero initialize $\{\boldsymbol{B}^\ell \in \mathbb{R}^{c_o^\ell \times r}\}_{\ell=1}^L$
3: Freeze pretrained weights $\{\boldsymbol{W}^\ell\}_{\ell=1}^L$
4: **for** $epoch = 1$ to $T$ **do**
5:     **for** each mini-batch $x \subset \mathcal{D}$ **do**
6:         **for** $\ell = 1$ to $L$ **do**
7:             Compute low-rank update:

$$\Delta\boldsymbol{W}^\ell \leftarrow \boldsymbol{B}^\ell \boldsymbol{A}^\ell$$

8:             Forward pass using $\boldsymbol{y} = (\boldsymbol{W}^\ell + \Delta\boldsymbol{W}^\ell)\boldsymbol{x}$
9:             Compute penalty terms for matrix $\boldsymbol{A}^\ell$ and $\boldsymbol{B}^{\ell\top}$ using Equation (7) and get $\mathcal{R}'_{K-rank}(\boldsymbol{A}^\ell)$ and $\mathcal{R}'_{K-rank}(\boldsymbol{B}^{\ell\top})$
10:         **end for**
11:         Compute task loss $\mathcal{L}_{task}$
12:         Compute the final loss function $\mathcal{L}_{\mathrm{total}} = \mathcal{L}_{\mathrm{task}} + \lambda \cdot \sum_{\ell=1}^L \left[ \mathcal{R}'_{\mathrm{K-rank}}(\boldsymbol{A}^\ell) + \mathcal{R}'_{\mathrm{K-rank}}(\boldsymbol{B}^{\ell\top}) \right]$
13:         Update $\{\boldsymbol{A}^\ell\}_{\ell=1}^L, \{\boldsymbol{B}^\ell\}_{\ell=1}^L$ by gradient descent:

$$\{\boldsymbol{A}^\ell \leftarrow \boldsymbol{A}^\ell - \eta\nabla_{\boldsymbol{A}^\ell}\mathcal{L}_{total}\}_{\ell=1}^L$$

$$\{\boldsymbol{B}^\ell \leftarrow \boldsymbol{B}^\ell - \eta\nabla_{\boldsymbol{B}^\ell}\mathcal{L}_{total}\}_{\ell=1}^L$$

14:     **end for**
15: **end for**
16: $\{\boldsymbol{W}_{out}^\ell \leftarrow \boldsymbol{W}^\ell + \boldsymbol{B}^\ell \boldsymbol{A}^\ell\}_{\ell=1}^L$
17: **return** $\{\boldsymbol{W}_{out}^\ell\}_{\ell=1}^L$

---

## E. Commonsense Reasoning Results with HiRA setting

Recall that in Table 1 in the main paper, we reproduce HiRA using the same setting as in DoRA, since the official code implementation of HiRA differs from that of DoRA and other competitors in many aspects. For example, HiRA selects the checkpoint with the best validation loss rather than the last checkpoint for inference. Also, HiRA uses the Seq2Seq trainer rather than the standard Huggingface trainer. Thus, it is unfair to directly compare our methods and other competitors with the official HiRA results.

In this section, we re-implement our ideas using the same setting as in HiRA and compare our methods with the official HiRA results for integrity. The experiments are conducted with LLaMA2-7B and LLaMA3-8B models on the commonsense reasoning datasets. The results in Table 5 show that we can surpass HiRA not only under the experimental setting of DoRA but also under the experimental setting of HiRA.

## F. More Ablation Studies

**LoKRA$^+$ without optimizing K-rank.** In the main paper, we replace the matrix multiplication operation with the Khatri-Rao product and optimize its K-rank to form our LoKRA$^+$. Since the K-rank is our core motivation, we do not present results from using the Khatri-Rao product alone without optimizing the K-rank. For integrity, we conduct an ablation study in the table below to show the results of using LoKRA$^+$ without optimizing the K-rank (i.e., equal to using the Khatri-Rao

*Table 5.* The results of the proposed LoKRA, LoKRA⁺, and HiRA with LLaMA2-7B and LLaMA3-8B on commonsense reasoning datasets under the HiRA experimental setting. For all metrics, higher is better.

| Model | Methods | Trainable Param. (%) | BoolQ | PIQA | SIQA | HellaS | WinoG | ARC-e | ARC-c | OBQA | Avg. |
|---|---|---|---|---|---|---|---|---|---|---|---|
| | HiRA ($r = 32$) | 0.83 | 71.2 | 83.4 | 79.5 | 88.1 | 84.0 | 86.7 | 73.8 | 84.6 | 81.4 |
| L2-7B | LoKRA ($r = 32$) | 0.83 | 73.7 | 84.5 | 80.3 | 89.2 | 85.4 | 88.6 | 75.9 | 85.8 | 82.9 |
| | LoKRA⁺ | 1.80 | 73.3 | 84.9 | 81.0 | 88.9 | 85.6 | 89.2 | 74.5 | 87.0 | 83.1 |
| | HiRA ($r = 32$) | 0.70 | 75.4 | 89.7 | 81.2 | 95.4 | 87.7 | 93.3 | 82.9 | 88.3 | 86.7 |
| L3-8B | LoKRA ($r = 32$) | 0.70 | 74.7 | 90.0 | 82.2 | 96.6 | 88.2 | 93.6 | 83.7 | 89.2 | 87.3 |
| | LoKRA⁺ | 1.52 | 76.3 | 90.4 | 82.5 | 95.8 | 88.6 | 93.9 | 84.5 | 89.0 | 87.6 |

product alone). Results on commonsense reasoning datasets with the LLaMA-7B model demonstrate the effectiveness of optimizing K-rank for the Khatri-Rao product.

*Table 6.* The results of LoKRA⁺ with and without optimizing K-rank on commonsense reasoning datasets with LLaMA-7B model. For all metrics, higher is better.

| Methods | BoolQ | PIQA | SIQA | HellaS | WinoG | ARC-e | ARC-c | OBQA | Avg. |
|---|---|---|---|---|---|---|---|---|---|
| LoKRA⁺ w/o K-rank | 69.6 | 82.2 | 78.2 | 84.8 | 79.0 | 82.2 | 64.7 | 77.4 | 77.3 |
| LoKRA⁺ | 70.5 | 83.0 | 79.5 | 86.0 | 80.5 | 82.9 | 68.5 | 81.0 | 79.0 |

**Hyper-parameter $\lambda$.** We give an ablation study on the hyper-parameter $\lambda$ in Equation (8). Experiments are conducted by fine-tuning LLaMA-7B with the proposed LoKRA and evaluated on the commonsense reasoning datasets. Figure 3 illustrates that when $\lambda = 0$, LoKRA degenerates to the original LoRA, and the best result is achieved at $\lambda = 1e - 3$. Our LoKRA can have better results than LoRA under a wide range of $\lambda$ (from 0 to $4e - 2$). We select the best $\lambda$ from $7e - 4$ to $7e - 3$ for other models.

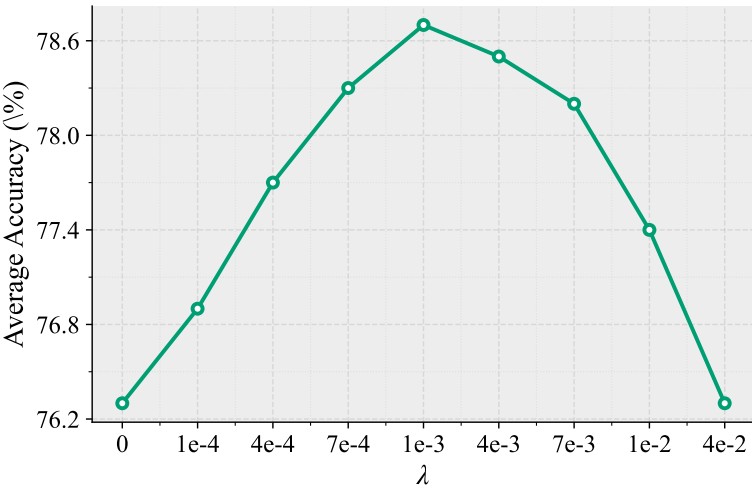

*Figure 3.* The results of the proposed LoKRA with LLaMA-7B on commonsense reasoning datasets using different hyper-parameter $\lambda$.

**Ablation on column/row-wise Khatri-Rao product in LoKRA⁺.** In Section 4.3, we decompose the output dimension $c_o$ into $pq$ to compute the Khatri-Rao product, and derive the lower bound on the column K-rank (corresponding to the input dimension) of the Khatri-Rao product. According to the analysis in Section 4.1, a higher column K-rank helps to preserve information on the input side, while a higher row K-rank enables richer and more distinguishable output signals. Therefore, it is natural to examine the effect of increasing the row K-rank that helps generate independent outputs.

Note that directly using the Khatri-Rao product results in a row K-rank with a lower bound of $1$. Thus, we define the row-wise Khatri-Rao product, which is also known as the face-splitting product or transposed Khatri-Rao product, as follows:

**Definition F.1** (row-wise Khatri-Rao product). Given two matrices $\boldsymbol{E} \in \mathbb{R}^{c_{out} \times q}$ and $\boldsymbol{F} \in \mathbb{R}^{c_{out} \times p}$, the row-wise Khatri-Rao product of $\boldsymbol{E}$ and $\boldsymbol{F}$ is defined as:

$$\boldsymbol{F} \overset{r}{\odot} \boldsymbol{E} = \begin{bmatrix} (\boldsymbol{f}_1^\top \otimes \boldsymbol{e}_1^\top)^\top \\ (\boldsymbol{f}_2^\top \otimes \boldsymbol{e}_2^\top)^\top \\ \vdots \\ (\boldsymbol{f}_{c_{out}}^\top \otimes \boldsymbol{e}_{c_{out}}^\top)^\top \end{bmatrix} \in \mathbb{R}^{c_{out} \times pq}$$

where $\overset{r}{\odot}$ stands for row-wise Khatri-Rao product, $\boldsymbol{e}_i$ ($\boldsymbol{f}_i$) is the $i$-th row of $\boldsymbol{E}$ ($\boldsymbol{F}$), and $\boldsymbol{f}_i \otimes \boldsymbol{e}_i$ is the Kronecker product which is already defined in Definition 4.2.

It is easy to derive the bound on the row K-rank of the row-wise Khatri-Rao product:

**Theorem F.2.** *Given two matrices $\boldsymbol{E} \in \mathbb{R}^{c_{out} \times q}$ and $\boldsymbol{F} \in \mathbb{R}^{c_{out} \times p}$, the row K-rank of the row-wise Khatri-Rao product result is bounded by:*

$$\min\{\mathrm{kr}_{\mathrm{row}}(\boldsymbol{E}) + \mathrm{kr}_{\mathrm{row}}(\boldsymbol{F}) - 1, c_{out}\} \leq \mathrm{kr}_{\mathrm{row}}(\boldsymbol{F} \overset{r}{\odot} \boldsymbol{E}) \leq c_{out}. \tag{23}$$

The proof is similar to Appendix B and is omitted here.

In the following, we conduct an ablation study to verify the effectiveness of using Khatri-Rao product and row-wise Khatri-Rao product in LoKRA$^+$ in Table 7. The experiments are conducted on the Commonsense Reasoning dataset using the LLaMA-7B model.

*Table 7.* The results of LoKRA$^+$ with Khatri-Rao product and row-wise Khatri-Rao product on commonsense reasoning datasets with LLaMA-7B model. For all metrics, higher is better.

| Methods | Commonsense Reasoning Datasets | | | | | | | | |
|---|---|---|---|---|---|---|---|---|---|
| | BoolQ | PIQA | SIQA | HellaS | WinoG | ARC-e | ARC-c | OBQA | Avg. |
| LoKRA$^+$ with Khatri-Rao | 70.5 | 83.0 | 79.5 | 86.0 | 80.5 | 82.9 | 68.5 | 81.0 | 79.0 |
| LoKRA$^+$ with row-wise Khatri-Rao | 70.3 | 83.2 | 79.4 | 84.6 | 81.8 | 82.7 | 67.1 | 82.0 | 78.9 |

The results show no significant difference between the Khatri–Rao product and the row-wise Khatri–Rao product. This is expected since the output of one layer in an LLM serves as the input to the subsequent layer (ignoring the activation function), making the optimization of column K-rank and row K-rank equally important.

Building on this, we next examine how optimizing only the column or row K-rank, rather than both, impacts the performance of LoKRA. In Section 4.2, we optimize the upper bound of the column K-rank and row K-rank of the update matrix $\Delta \boldsymbol{W}$. Here, we conduct an ablation study to isolate the effect of optimizing each component individually.

*Table 8.* The results of LoKRA with only column/row K-rank optimized on commonsense reasoning datasets with LLaMA-7B model. For all metrics, higher is better.

| Methods | Commonsense Reasoning Datasets | | | | | | | | |
|---|---|---|---|---|---|---|---|---|---|
| | BoolQ | PIQA | SIQA | HellaS | WinoG | ARC-e | ARC-c | OBQA | Avg. |
| LoRA | 67.5 | 80.8 | 78.2 | 83.4 | 80.4 | 78.0 | 62.6 | 79.1 | 76.3 |
| LoKRA with only column K-rank optimized | 67.8 | 82.4 | 78.9 | 86.6 | 81.1 | 80.0 | 67.4 | 81.6 | 78.2 |
| LoKRA with only row K-rank optimized | 69.1 | 82.4 | 79.1 | 85.8 | 81.4 | 81.2 | 66.4 | 80.6 | 78.3 |
| LoKRA | 69.8 | 82.2 | 78.1 | 87.5 | 80.5 | 82.3 | 67.2 | 82.0 | 78.7 |

We observe that optimizing only the column K-rank or the row K-rank improves performance over LoRA, whereas optimizing both yields the best results.

# G. Limitations

**Limitations of LoKRA.** Although we simplify the calculation of the penalty term from Equation (6) to Equation (7), LoKRA still introduces additional computational overhead during training compared to the original LoRA method, resulting in longer training time.

**Limitations of LoKRA$^+$.** While LoKRA$^+$ achieves better performance than LoKRA, it introduces more trainable parameters and consequently longer training time. Besides, the number of trainable parameters cannot be flexibly adjusted as LoRA does, since the constraint $pq = c_{out}$ must be satisfied, as shown in Section 4.3 in the main paper. Finally, LoKRA$^+$ is less suitable when $c_{out}$ is a prime number or admits only highly unbalanced factors. In such cases, the structure of the Khatri-Rao product is restricted, limiting the trainable parameter efficiency.

Overall, our analysis is limited to theoretically verifying the compatibility of K-rank with matrix multiplication and the Khatri-Rao product. Exploring matrix operations that offer more flexible parameter control while maintaining solid theoretical guarantees for K-rank is an interesting direction for future work.

