# OpenReview forum: "Low Kruskal-Rank Adaptation"
_ICML.cc/2026/Conference — ICML 2026 regular_

### Official Review · Reviewer_RG4J · 2026-02-15

**Soundness:** 2
**Presentation:** 3
**Significance:** 2
**Originality:** 2
**Overall Recommendation:** 3
**Confidence:** 3

**Summary:**

This paper revisits the rank constraint in LoRA and argues that conventional matrix rank is insufficient to capture redundancy and duplicated update directions. To address this limitation, the authors introduce LoKRA, which promotes higher Kruskal rank in update matrices via a log-determinant regularization that encourages stronger directional independence. They further propose LoKRA+, replacing matrix multiplication with the Khatri-Rao product to obtain tighter theoretical guarantees on Kruskal rank. Extensive experiments on commonsense and mathematical reasoning tasks show that LoKRA and LoKRA+ consistently outperform LoRA and other PEFT baselines under comparable parameter budgets, demonstrating improved expressiveness and empirical performance while maintaining parameter efficiency.

**Compliance With Llm Reviewing Policy:**

Affirmed.

**Key Questions For Authors:**

See Weaknesses.

**Limitations:**

yes

**Strengths And Weaknesses:**

- Strengths
1. `More expressive update criterion`. The paper identifies a key limitation of standard LoRA and introduces Kruskal rank as a stricter measure of directional independence. By promoting higher Kruskal rank, the method effectively reduces redundancy and duplicated update directions, enhancing expressive capacity under the same parameter budget.
2. `Strong theoretical grounding`. The proposed LoKRA and LoKRA+ frameworks are supported by formal upper and lower bounds on Kruskal rank. The log-determinant regularization and the Khatri-Rao reparameterization (in LoKRA+) provide principled guarantees that better align optimization with update diversity.
3. `Consistent empirical gains across settings`. Extensive experiments on commonsense and mathematical reasoning tasks demonstrate consistent improvements over LoRA and prior PEFT baselines, in both supervised fine-tuning and reinforcement learning scenarios, while maintaining parameter efficiency.

- Weaknesses
1. The categorization of follow-up studies on LoRA in the introduction seems somewhat inappropriate. For example, it overlooks works on parameter initialization[1-2] and architecture-evolution variants[3-5].
2. Concerns about the motivation. The paper claims that the rank of standard LoRA fails to identify redundancy and duplicated directions in parameter updates. However, LoRA has already been widely accepted in practice. It remains unclear how severe these rank-related limitations are in real settings, especially given LoRA's simplicity. In contrast, many more complex methods are often difficult to deploy and sometimes yield marginal gains or even performance degradation.
3. Considering the rapid iteration of LLMs, could the authors conduct experiments on more recent base models ($e.g.$, Qwen3-8B) (experiments on base models such as LLaMA-7B and LLaMA2-7B are arguably unnecessary in the main paper)?  It would also be valuable to include MoE+LoRA as a baseline for comparison. In addition, experiments are recommended to be organized by different domains, rather than focusing primarily on math-related tasks.
4. Could the authors open-source their code to facilitate further review and a deeper understanding of the proposed method?


[1] PiSSA: Principal Singular Values and Singular Vectors Adaptation of Large Language Models
[2] LoRA-Pro: Are Low-Rank Adapters Properly Optimized
[3] When MOE Meets LLMs: Parameter Efficient Fine-tuning for Multi-task Medical Applications
[4] HydraLoRA: An Asymmetric LoRA Architecture for Efficient Fine-Tuning
[5] CoLA: Collaborative Low-Rank Adaptation

---

> ### Author Rebuttal · Authors · 2026-03-31
>
> Thank you for the constructive critique and for highlighting the importance of testing on recent baselines. We provide our clarifications below.
>
> ---
>
> **W1: Overlooked literature.**
>
> **Re W1:** Due to the vast literature on LoRA, our initial submission focused mostly on rank-related works. We appreciate you pointing out these important dimensions. We will cite and discuss the parameter initialization [1-2] and architecture evolution [3-5] papers in the revised related work section.
>
> ---
>
> **W2: Motivation and complexity. Are rank limitations in LoRA severe enough in practice?**
>
> **Re W2:** We fully agree that LoRA is a simple and widely adopted baseline, but its availability does not mean it is free from structural flaws. Our theoretical analysis reveals its inherent inability to penalize highly correlated or duplicated update directions.
>
> Empirically, these rank limitations and redundancy issues in LoRA can be surprisingly severe in practice. For example, as shown in **Table 1** in the main paper, simply increasing the rank of standard LoRA to gain more capacity paradoxically leads to severe performance degradation. On LLaMA3-8B, increasing LoRA's rank from $r=32$ to $r=64$ causes the average accuracy to drop drastically from 80.9% to 75.1%. In extreme cases (e.g., LLaMA-7B LoRA $r=64$), the performance on the HellaS benchmark completely collapses to 17.6%. In contrast, our LoKRA mitigates this structural redundancy and robustly achieves up to 86.3% average accuracy on LLaMA3-8B without suffering from such collapse.
>
> Furthermore, optimizing this structural flaw does not sacrifice simplicity. LoKRA maintains the standard LoRA training and inference pipeline and only adds a loss regularizer (implemented in ~20 lines of PyTorch code). LoKRA+ simply replaces matrix multiplication with the Khatri-Rao product, which requires just one line of code (using `torch.einsum`). Therefore, our methods deliver significant and robust performance gains while remaining easy to deploy.
>
> ---
>
> **W3: Experiments on recent base models (Qwen3-8B), MoE+LoRA, and focusing primarily on math-related tasks.**
>
> **Re W3:** We conducted new experiments on the latest Qwen3-8B-Base model for commonsense reasoning. As shown below, LoKRA and LoKRA+ consistently outperform both standard LoRA and DoRA across different settings, demonstrating our generalizability to wide model architectures.
>
> | PEFT Method | # Params (%) | BoolQ | PIQA | SIQA | HellaSwag | WinoGrande | ARC-e | ARC-c | OBQA | Avg |
> | - | - | - | - | - | - | - | - | - | - | - |
> | LoRA (r=32) | 0.71 | 73.43 | 89.17 | 80.14 | 83.41 | 85.40 | 96.04 | 87.80 | 89.60 | 85.62 |
> | LoRA (r=64) | 1.42 | 71.41 | 87.92 | 79.53 | 93.62 | 83.82 | 94.57 | 86.26 | 90.80 | 85.99 |
> | DoRA (r=32) | 0.72 | 73.33 | 89.28 | 80.45 | 94.25 | 85.63 | 95.66 | 88.82 | 91.40 | 87.35 |
> | DoRA (r=64) | 1.43 | 73.73 | 89.28 | 80.25 | 94.51 | 86.66 | 95.88 | 89.25 | 92.60 | 87.77 |
> | LoKRA (r=32) | 0.71 | 74.16 | 90.53 | 81.63 | 95.27 | 86.90 | 97.22 | 92.58 | 92.80 | **88.89** |
> | LoKRA+ | 1.53 | 75.90 | 90.86 | 82.14 | 95.77 | 86.66 | 97.22 | 91.81 | 92.80 | **89.15** |
>
> Furthermore, to validate our approach on MoE architectures, we evaluated the Qwen3-30B-A3B-Base model. Given the massive scale of this model and the limited rebuttal time, we trained for 1 epoch and controlled for a closely matched trainable parameter budget across methods.
>
> | Method | # Params (%) | BoolQ | PIQA | SIQA | HellaSwag | WinoGrande | ARC-e | ARC-c | OBQA | Avg |
> | - | - | - | - | - | - | - | - | - | - | - |
> | LoRA (r=32)| 3.55 | 71.62 | 91.84 | 80.50 | 94.75 | 85.32 | 98.48 | 94.71 | 94.40 | 88.95 |
> | LoKRA (r=32)| 3.55 | 75.96 | 92.95 | 82.81 | 96.45 | 90.21 | 98.65 | 95.14 | 96.20 | **91.04** |
> | LoKRA+ | 3.72 | 77.86 | 93.91 | 83.11 | 96.54 | 91.08 | 98.86 | 95.47 | 96.60 | **91.67** |
>
> Even under the restricted 1-epoch training setting on an MoE backbone, LoKRA (+2.09%) and LoKRA+ (+2.72%) demonstrate substantial and consistent gains over standard LoRA. This strongly confirms that addressing Kruskal-rank limitations provides robust improvements independent of the underlying architecture (Dense or MoE). We will include all these new results in the final version of the paper.
>
> Finally, rather than focusing primarily on math-related tasks in Table 2, in Table 1 we conduct experiments on the commonsense reasoning datasets, which include reading comprehension (BoolQ), physical commonsense (PIQA), social commonsense (SIQA), grounded commonsense inference (HellaSwag), coreference/common-sense disambiguation (WinoGrande), and science QA benchmarks (ARC-E/ARC-C/OBQA).
>
> ---
>
> **W4: Open-sourcing code.**
>
> **Re W4:** We are committed to reproducibility and will fully open-source our code and training scripts upon the final decision of this paper being made.
>
> ---
>
> We hope that our clarifications can address your concerns, and we remain happy to provide further detailed discussions.
>
> Best,
>
> Authors 3243

---

> > ### Author Rebuttal · Reviewer_RG4J · 2026-04-02
> >
> > I look forward to the revised paper using the latest base model in the experiments.

---

> > > ### Author Response · Authors · 2026-04-03
> > >
> > > Thank you for raising your score. We will include results on the latest base model in the final version of the paper.
> > >
> > > Best,
> > >
> > > Authors 3243

---

### Official Review · Reviewer_L1mZ · 2026-02-16

**Soundness:** 3
**Presentation:** 2
**Significance:** 1
**Originality:** 2
**Overall Recommendation:** 4
**Confidence:** 1

**Summary:**

The authors propose using Kruskal rank (K-rank) as a stricter notion of independence and introduce Low Kruskal Rank Adaptation (LoKRA), which adds a penalty that encourages higher K-rank by maximizing a log-determinant objective over submatrices of the LoRA factors.

**Compliance With Llm Reviewing Policy:**

Affirmed.

**Key Questions For Authors:**

What is the exact sampling scheme for choosing the column/row subset (e.g., uniform over all subsets, uniform over indices without replacement, fixed vs. re-sampled per step, per-layer vs. shared)?

How sensitive are results to the stochastic choices in the approximation (both the subset selection and the randomly sampled subset size)? Do you observe meaningful variance across random seeds?

Beyond adding a small diagonal term, what is the exact numerical method used to compute the log-determinant during training, and what safeguards are used if the matrix is nearly singular or produces NaNs?

Are all regularizer settings, including the penalty weight, kept identical across all layers and all adapter matrices, or do you use layer-wise scaling? Also, are these hyperparameters tuned per dataset/model, or fixed once and reused?

**Limitations:**

Yes

**Strengths And Weaknesses:**

Strengths:
The paper provides a clear motivation for why matrix rank can miss redundancy in update directions and why Kruskal rank is a more stringent criterion for “global” independence. The core methods are accompanied by a theoretical analysis of K-rank behavior under the proposed operations. The authors include ablation studies that help attribute gains to the proposed components. The paper acknowledges and partially quantifies compute overhead, e.g., training-time alignment experiments and discussion of why removing certain simplifications is infeasible.

Weaknesses:
- The diversity/K-rank regularizer is computed on randomly sampled subsets (and sampled subset sizes) rather than the full objective. This approximation can be sensitive to sampling choices and weakens how cleanly the intended guarantee carries over in practice.
- LoKRA+ enforces a hard coupling between adapter dimensions and then selects a specific factor shape to fit it. This reduces flexibility v.s. standard LoRA and can create edge cases when the layer dimension doesn’t factor nicely, limiting applicability across architectures/layers.
- Key reproduction details are underspecified, such as how subsets are chosen for the log-det penalty, how numerical stability is ensured for log-det computations, and how hyperparameters are set across layers and model scales.

---

> ### Author Rebuttal · Authors · 2026-03-31
>
> We are grateful for your detailed review and insightful questions regarding our implementation. We address them below.
>
> ---
>
> **W1, Q1, Q2: Stochastic choices in the approximation, sampling scheme, with or without replacement, and variance.**
>
> **Re W1, Q1, Q2:** Thanks for pointing this out. As computing the exact Kruskal rank is NP-hard (requiring the determinant of $10^{80}$ submatrices as shown in Line 421, column right in the main paper), we use a randomly sampled sub-matrix at each step to provide a Monte Carlo estimate of the full objective, conceptually similar to how minibatch SGD approximates full gradient descent.
>
> **Sampling Scheme:** For a given low-rank matrix, we first randomly sample the target sub-matrix size $k' \sim \text{Uniform}${$2, \dots, k$}. We then uniformly sample $k'$ columns/rows from the original matrix and form one sub-matrix to evaluate the penalty. Importantly, this sampling is performed **with replacement** across different training steps and is independent across all layers and matrices.
>
> **Variance:** We evaluated the sensitivity to this stochasticity across 5 random seeds on LLaMA3-8B. The results show that our method remains robust and stable, with variance comparable to standard LoRA.
>
> |Method|Mean Acc.|Std.|
> |-|-|-|
> |LoRA|80.75|0.19|
> |LoKRA|85.83|0.14|
> |LoKRA+|86.27|0.15|
>
> ---
>
> **W2: Hard coupling of adapter dimensions in LoKRA+ and edge cases.**
>
> **Re W2:** We completely agree that this is a structural limitation of LoKRA+, as discussed in Appendix G. LoKRA+ is indeed less flexible if $c_{out}$ is a prime number. Fortunately, the hidden dimensions of almost all popular LLMs (e.g., 4096, 11008) are highly composite numbers that can be easily factorized into two balanced integers, preventing practical deployment issues in most scenarios.
>
> ---
>
> **W3 & Q3: Numerical stability of log-det computation. Hyperparameters.**
>
> **Re W3 & Q3:** For the hyper-parameter setting, we mentioned that we use the same setting as DoRA, except for the K-rank loss in lines 321-323, column right. The experiment of the choice of hyper-parameter $\lambda$ is given in Appendix F.
>
> The subset chosen method is specified in previous. To compute the log-determinant safely, we calculate `gram_A = A_sub @ A_sub.T` and apply `torch.slogdet(gram_A + 1e-6 * torch.eye(k'))`. If the matrix is nearly singular at the very beginning of training and yields NaNs or extreme values, we safely catch the exception and skip the K-rank loss update for that specific step. This issue almost never occurs once training stabilizes.
>
> ---
>
> **Q4: Hyperparameter scaling across layers.**
>
> **Re Q4:** We maintain the exact same regularizer setting and penalty weight ($\lambda$) across all layers and adapter matrices. $\lambda$ is tuned once per model architecture from {$\{7e-4, 1e-3, 4e-3, 7e-3\}$} (as detailed in Appendix F) and remains fixed across all downstream datasets.
>
> ---
>
> We hope our explanations sufficiently address your comments. We welcome any additional feedback.
>
> Best,
>
> Authors 3243

---

> > ### Author Rebuttal · Reviewer_L1mZ · 2026-04-05
> >
> > Thanks authors for addressing my concerns in their rebuttal. I’ll keep my already positive review.

---

> > > ### Author Response · Authors · 2026-04-05
> > >
> > > Thank you for your support.
> > >
> > > Best,
> > >
> > > Authors 3243

---

### Official Review · Reviewer_Y1B1 · 2026-02-20

**Soundness:** 3
**Presentation:** 3
**Significance:** 3
**Originality:** 3
**Overall Recommendation:** 4
**Confidence:** 4

**Summary:**

This paper aims to improve the Kruskal rank of LoRA adapter, so that redundant update directions in the update space can be avoided. This is achieved by regularizing the volume of the polyhedron spanned by the columns (i.e., log-determinant of Gram matrix) with two simplifications, which results in the LoKRA method. To further increase the lower bound of $kr(\Delta W)$, Khatri-Rao product is leveraged in LoKRA+ to parameterize the weight update, leading to increased parameters yet enhanced performance.

**Compliance With Llm Reviewing Policy:**

Affirmed.

**Final Justification:**

I keep my positive score of 4.

**Key Questions For Authors:**

See weaknesses.

**Limitations:**

Yes

**Strengths And Weaknesses:**

**Strengths**
1. The motivation for improving the Kruskal rank is clear, and the paper is well-written.
2. While calculating Kruskal rank is an NP-hard combinatorial problem, a tractable alternative regularization is provided (7) to simplify the computation.
3. Experimental results are promising, showcasing a consistent improvement over other baselines under different setups.

**Weaknesses**
1. The Khatri-Rao product in LoKRA+ is not compute- and memory-efficient, as (11) requires computing and backpropagating the $c_{out} \times c_{in}$ matrix $D \odot C$. While standard LoRA is performed via $B(Ax)$. This growth of computational cost and memory footprint can be potentially alleviated by certain parameterizations; see an example in [1]. It would be helpful if the authors can provide some insights regarding how to mitigate this scalability issue.
2. Related approaches including Kronecker-product-based method are mentioned in the introduction but excluded from the experiments, and LoRA variants are missing from the math reasoning tests in Table 2.
3. A typo in line 271, left column: $\delta W$ should be $\Delta W$.

[1] ABBA-Adapters: Efficient and Expressive Fine-Tuning of Foundation Models.

---

> ### Author Rebuttal · Authors · 2026-03-31
>
> Thank you for the valuable suggestions and for pointing out the scalability considerations. Our responses are as follows.
>
> ---
>
> **W1: Compute and memory inefficiency of Khatri-Rao product in LoKRA+ / Potential mitigations (e.g., ABBA-Adapters).**
>
> We agree that the Khatri-Rao product naturally increases computational overhead compared to standard matrix multiplication. This is a trade-off for higher model capacity and performance. To mitigate this scalability issue, several strategies can be employed:
>
> 1. **Execution-Level Optimization:** Mathematically, the forward pass $y = (D \odot C)x$ can be decoupled via tensor contraction: $y = \text{flatten}( (x \circ C) D^T )$, where $\circ$ denotes row-wise broadcasting. This elegantly avoids instantiating the massive $\Delta W \in \mathbb{R}^{c_{out} \times c_{in}}$ matrix in memory and theoretically reduces FLOPs.
> In practice, however, we found that this implementation requires an element-wise broadcasting operation, and needs more training time and GPU memory compared to directly computing the original form of Khatri-Rao product, as shown in the table below.
>
> | LoKRA+ method| Training Time / Step (s) | GPU Memory (GB) |
> | - | - | - |
> | original Khatri-Rao | 1.43 | 50.11 |
> | Execution-Level Optimization| 3.20| 66.02|
>
> 2. **Further Factorization (e.g., ABBA-Adapters):** As you insightfully suggested, we can further parameterize the factors $D$ and $C$ into lower-rank structures (e.g., $D = D_1 D_2$). This structural re-parameterization would fundamentally reduce trainable parameters and computational complexity.
>
> 3. **Selective Application:** We can apply the Khatri-Rao product only to critical layers while using standard LoRA for others.
>
> 4. **Group-wise Khatri-Rao product:** For each row of the standard Khatri-Rao product, we derive $d_i \otimes c_i\in\mathbb R^{pq}$ in Eq.10. Group-wise Khatri-Rao product can divide $d_i$ and $c_i$ into $g$ groups, and compute the operation within the group, and finally reduce the row dimension from $\mathbb R^{pq}$ to $\mathbb R^{pq/g}$, therefore reducing the computational overhead.
>
> Except for method 1 mentioned above (mathematically equivalent to the original form), the other methods may impact the model performance, and need careful theoretical analysis to find out whether they still have reasonable upper and lower K-rank bounds. We will add the above discussion to the final paper.
>
> ---
>
> **W2: Missing comparisons with Kronecker-product methods and LoRA variants on math reasoning.**
>
> **Re W2:** We compare the official results of Kron-LoRA with our methods below. While Kron-LoRA has fewer trainable parameters (as it factorizes both row and column dimensions), it yields noticeably worse performance.
>
> | Model | Methods | Param (%) | BoolQ | PIQA | SIQA | HellaS | WinoG | ARC-e | ARC-c | OBQA | Avg. |
> | - | - | - | - | - | - | - | - | - | - | - | - |
> | LLaMA2-7B | Kron-LoRA | 0.05 | 80.5 | 82.1 | 61.1 | 86.8 | 73.2 | 81.4 | 57.9 | 67.4 | 73.8 |
> | - | LoKRA | 0.83 | 72.8 | 83.8 | 80.3 | 89.5 | 82.8 | 85.0 | 72.1 | 82.4 | **81.1** |
> | - | LoKRA+ | 1.80 | 72.5 | 84.6 | 79.5 | 90.6 | 84.1 | 85.7 | 73.0 | 83.4 | **81.7** |
> |LLaMA3-8B|Kron-LoRA|0.05|84.3|84.0|59.3|89.4|69.9|77.2|55.5|68.2|73.5|
> |-|LoKRA|0.70|74.9|89.1|81.3|95.5|86.9|90.5|81.2|87.6|**85.9**|
> |-|LoKRA+|1.52|76.0|88.5|81.1|95.7|86.0|81.9|81.6|89.6|**86.3**|
>
> Regarding math reasoning, RL fine-tuning on pipelines like DeepScaleR and STILL takes dozens of days to complete. Due to the limited rebuttal period, we were unable to run LoRA variants on all pipelines. However, we evaluated DoRA on the Open-RS1 pipeline below, which shows no obvious superiority to LoRA and falls behind our methods.
>
> | Pipeline | Methods | AIME24 | AIME25 | AMC23 | MATH500 | GPQA | Minerva | Avg. |
> | - | - | - | - | - | - | - | - | - |
> | Open-RS1 | Tina (LoRA) | 43.33 | 20.00 | 80.00 | 84.00 | 35.35 | 28.68 | 48.56 |
> | - | DoRA | 40.00 | 26.67 | 72.50 | 84.40 | 41.41 | 28.68 | 48.94 |
> | - | LoKRA | 40.00 | 30.00 | 80.00 | 85.40 | 33.84 | 31.62 | **50.14** |
> | - | LoKRA+ | 43.33 | 26.67 | 82.50 | 85.40 | 38.89 | 31.99 | **51.46** |
>
> ---
>
> **W3: Typo in line 271.**
>
> **Re W3:** Thank you. We have fixed $\delta W$ to $\Delta W$.
>
> ---
>
>
> We hope the above clarifications address your concerns. Please let us know if further clarification is needed.
>
> Best,
>
> Authors 3243

---

> > ### Author Rebuttal · Reviewer_Y1B1 · 2026-04-02
> >
> > Thank you for the detailed explanation and additional experiments. My concerns are adequately addressed. I keep my positive score.

---

> > > ### Author Response · Authors · 2026-04-03
> > >
> > > Thank you for keeping your positive score.
> > >
> > > Best,
> > >
> > > Authors 3243

---

### Official Review · Reviewer_YDVn · 2026-03-13

**Soundness:** 3
**Presentation:** 4
**Significance:** 3
**Originality:** 3
**Overall Recommendation:** 5
**Confidence:** 3

**Summary:**

The paper aims at increaining  the expressive diversity of LoRA-style updates an proposes a LoRA variant moving the usual emphasis on standard rank to Kruskal rank as the relevant notion of “update with diversity". Indeed, Kruskal rank penalizes duplicated or overly correlated directions in the update matrix. Their method adds a log-determinant regularizer on randomly sampled submatrices of the LoRA factors, with the goal of encouraging more independent columns/rows during training. They also propose LoKRA+, which swaps the standard matrix product for a Khatri–Rao product, motivated by a lower-bound result on the Kruskal rank of the resulting update. Empirically, they report moderate but fairly consistent gains over LoRA and several PEFT baselines on commonsense reasoning with LLaMA-family models and on math reasoning with DeepSeek-family models under RL fine-tuning.

**Compliance With Llm Reviewing Policy:**

Affirmed.

**Final Justification:**

I was already positive about the initial submission, and the authors’ rebuttal has further strengthened my favorable assessment.

**Key Questions For Authors:**

See **Weaknesses** here above.

In particular,
* In the experimental  section, could you report results on the actual Kruskal rank, approximate Kruskal rank,  dependence patterns, or any robust proxy over training?
* As the paper simplifies the full initial objective quite aggressively, it it possible to have some further ablations quantifying the effects of these strong simplifications?
* In Table 1, do you have any idea of the weird strong drop of performance of HellaS result for LLaMA-7B LoRA r=64?
* Could you perform a short cost analysis in terms of compute time, memory footpring, and scaling with model width (in particular r)?

**Limitations:**

Yes, he authors adequately discussed the limitations and potential negative societal impact of their work (even if very briefly).

**Strengths And Weaknesses:**

**Strengths**

* The main ide of the paper, namely to depart from the usual “rank = expressive power” concept towards the Kruskal Rank - which is a more global measure of expressivity/independance/diversity in the update directions, is clearly and intuitively explained.

* The theoretical part (upper bounds for the Kruskal rank of $BA$ product, and the lower-bound story for the Khatri–Rao product)  is also well structured and easy to understand: this gives a very coherent mathematical rationale for the underlying methods.

* Empirically, the method appears consistently competitive. Even if hhe gains are not big, they are there across several model families.

**Novelty**
* The novel part is the explicit use of Kruskal rank as the structuring principle for LoRA adaptation, with a concrete regularizer and the associated bounds. The use of Khatri-Rao product in LORA updates is not new. For instance,  KRAdapter (ICCV 2025 paper) explicitly motivates Khatri–Rao updates as a way to produce higher-rank, more expressive fine-tuning updates.  So, going from LOKRA to LOKRA++ is only more an incremental refinement.

**Weaknesses**
* There is some discrepancy between the theoretical part of the paper and its experimental part. While the paper argues that higher Kruskal rank reduces redundancy and is the real reason for better adaptation, the experimental results are not measuring these effects directly. Indeed, the experiments mostly show task accuracy, not direct evidence that the learned updates actually have better Kruskal-rank-related structure.
* There are no confidence intervals or multi-seed variance reports in the main results tables. As the improvements over baselines are sometimes small, this would have helped to better appreciate the real gains of the method.

---

> ### Author Rebuttal · Authors · 2026-03-31
>
> We appreciate your positive assessment and constructive feedback. Please find our detailed responses below.
>
> ---
>
> **W1 & Q1: Discrepancy between theory and experiments / Reporting actual Kruskal rank.**
>
> **Re W1 & Q1:** Computing the exact Kruskal rank for a given matrix is an NP-hard problem. Thus, it is computationally intractable to derive the actual Kruskal rank during training. For instance, evaluating the full Kruskal rank of a single parameter matrix in LLaMA-7B would require computing the determinant of $10^{80}$ submatrices (as mentioned in Line 421 in the main paper, column right).
>
> However, the simplified Kruskal rank loss (Eq. 7) provides a tractable and robust surrogate: at each step, we estimate the full objective via a randomly sampled sub-matrix, akin to how minibatch SGD approximates full gradient descent. In the table below, we show this loss averaged over all low-rank matrices in LLaMA-7B. It significantly decreases after training, demonstrating that our method effectively optimizes the Kruskal-rank-related structure. We will include these training curves in the final version.
>
> | Metric | Before Training | After Training |
> | - | - | - |
> | Simplified K-rank Loss | 34.5 | 15.9 (-53.9%) |
>
> ---
>
> **W2: Lack of confidence intervals / multi-seed variance.**
>
> **Re W2:** Thank you for the suggestion. We repeated the commonsense reasoning experiments on LLaMA3-8B with 5 different random seeds. The mean and standard deviation are shown below. The improvements of LoKRA and LoKRA+ over LoRA are consistent and statistically significant, with comparable stability.
>
> | Method | Mean Acc. (%) | Std. |
> | - | - | - |
> | LoRA | 80.75 | 0.19 |
> | LoKRA | 85.83 | 0.14 |
> | LoKRA+ | 86.27 | 0.15 |
>
> ---
>
> **Q2: Further ablations on simplifications.**
>
> **Re Q2:** We provided the corresponding ablation study in Table 4 in the main paper. As discussed in Lines 410-422 (column right), there are two simplifications. We ablated the first one in Table 4. The second simplification (using randomly sampled sub-matrices instead of all sub-matrices) cannot be ablated or removed because computing the full objective is NP-hard, as explained above.
>
> ---
>
> **Q3: Performance drop of HellaS for LLaMA-7B LoRA r=64.**
>
> **Re Q3:** This is an official result directly derived from DoRA (https://github.com/NVlabs/DoRA/tree/main/commonsense_reasoning). During our reimplementation, we also observed this performance drop with approximately a 40% probability. We hypothesize that LLaMA-7B is particularly sensitive to redundant information introduced by higher ranks, as we do not observe such degradation on other models.
>
> ---
>
> **Q4: Cost analysis (compute, memory, scaling).**
>
> **Re Q4:** The table below shows the average training time per step and GPU memory cost on LLaMA3-8B. LoKRA introduces a slight computational overhead due to the K-rank loss. LoKRA+ consumes more resources due to the Khatri-Rao product. However, as shown in Figure 2 and Table 3 in the main paper, when aligning the total training time, LoKRA+ still achieves the best performance. We have also discussed these resource limitations in Appendix G.
>
> | Method | r | Training Time / Step (s) | GPU Memory (GB) |
> | - | - | - | - |
> | LoRA | 32 | 0.69 | 31.03 |
> | LoKRA | 32 | 0.86 | 39.64 |
> | LoRA | 64 | 0.90 | 32.11 |
> | LoKRA | 64 | 1.06 | 41.42 |
> | LoKRA+ | - | 1.43 | 50.11 |
>
> ---
>
> We hope the above clarifications adequately address your concerns. We would be happy to provide further details if needed.
>
> Best,
>
> Authors 3243

---

> > ### Author Rebuttal · Reviewer_YDVn · 2026-04-03
> >
> > My main concerns appear to be adequately addressed.Thanks for the detailed clarifications and additional results!

---

> > > ### Author Response · Authors · 2026-04-04
> > >
> > > Thanks for your support.
> > >
> > > Best,
> > >
> > > Authors 3243

---

### Decision · Program_Chairs · 2026-04-30

**Decision:**

Accept (regular)

**Comment:**

Motivated by identifying a structural limitation in standard LoRA that the standard matrix rank fails to adequately penalize duplicated directions and redundancy in the update subspace, this paper proposes LoKRA, which uses a log-determinant regularizer on sampled submatrices to encourage higher Kruskal rank, thereby promoting update diversity. The initial reviews were generally positive, with lowest being one weak reject. Reviewers uniformly praised the paper's clear motivation, strong theoretical grounding, and consistent empirical gains. However, reviewers raised several valid concerns during the initial phase such as computational and memory overhead, instability of the random sub-matrix sampling used to approximate the NP-hard Kruskal rank computation, baselines such as as comparison to Kronecker-product methods, etc. The authors provided a  comprehensive and effective rebuttal. Overall, the paper makes a solid contribution to the highly relevant sub-field of PEFT.  The authors are encouraged to include all the additional experiments provided in the rebuttal (specifically the 5-seed variance tables, the Qwen3-8B and MoE results, the Kron-LoRA comparisons, and the expanded discussion on computational mitigations)  into the subsequent version of the paper to improve its quality.